# S-Benproperine, an Active Stereoisomer of Benproperine, Suppresses Cancer Migration and Tumor Metastasis by Targeting ARPC2

**DOI:** 10.3390/ph15121462

**Published:** 2022-11-25

**Authors:** Hyun-Jin Jang, Yae Jin Yoon, Jiyeon Choi, Yu-Jin Lee, Sangku Lee, Wansang Cho, Wan Gi Byun, Seung Bum Park, Dong Cho Han, Byoung-Mog Kwon

**Affiliations:** 1Laboratory of Chemical Biology and Genomics, Korea Research Institute of Bioscience and Biotechnology, 125 Gwahakro, Daejeon 34141, Republic of Korea; 2Department of Chemistry, Seoul National University, Seoul 08826, Republic of Korea; 3KRIBB School of Bioscience, University of Science and Technology in Korea, 217 Gajeongro, Daejeon 34113, Republic of Korea; 4Central Research Institute, VS Pharm Tech Co., Ltd., Daejeon 35209, Republic of Korea

**Keywords:** stereoisomers, benproperine, metastasis, actin-related protein 2/3 complex, actin-related protein 2/3 complex subunit 2

## Abstract

Metastasis, in which cancer cells migrate to other tissues and form new tumors, is a major cause of both cancer death and treatment failure. In a previous study, benproperine (Benp) was identified as a cancer cell migration inhibitor and an inhibitor of actin-related protein 2/3 complex subunit 2 (ARPC2). However, Benp is a racemic mixture, and which stereoisomer is the active isomer remains unclear. In this study, we found that S-Benp is an active isomer and inhibits the migration and invasion of cancer cells much more strongly than R-Benp, with no effect on normal cells. The metastasis inhibitory effect of S-Benp was also verified in an animal model. Validating that inhibitors bind to their targets in cells and tissues has been a very challenging task in drug discovery. The direct interactions between ARPC2 and S-Benp were verified by surface plasmon resonance analysis (SPR), a cellular thermal shift assay (CETSA), and drug affinity responsive target stability (DARTS). In the mutant study with ARPC2^F225A^ cells, S-Benp did not bind to ARPC2^F225A^ according to CETSA and DARTS. Furthermore, we validated that S-Benp colocalized with ARPC2 in cancer cells and directly bound to ARPC2 in tumor tissues using Cy3-conjugated S-Benp according to CETSA. Finally, actin polymerization assays and immunocytochemistry showed that S-Benp suppressed actin remodeling such as lamellipodium formation. Taken together, our data suggest that S-Benp is an active stereoisomer of Benp and a potential metastasis inhibitor via ARPC2 binding.

## 1. Introduction

In 2022, the American Cancer Society predicted that approximately 2 million new people will develop cancer and that approximately 610,000 people will die of cancer in the United States [1]. Previous studies indicated that metastasis causes the most cancer deaths and treatment failures [2,3]. Metastatic cancer is difficult to remove by general surgery and chemotherapies since cancer cells spread to other organs and obtain drug resistance during metastasis [3]. However, the majority of anticancer research has focused on primary cancer cell death rather than metastasis inhibition [4].

Through metastasis, some cancer cells escape from the primary tumor and generate a secondary niche [5]. During invasion and migration, which are the initial steps of cancer metastasis, cancer cells change their properties and adapt to different microenvironments [5,6]. Actin reorganization, as an essential step for cell migration, increases the invasiveness of cancer cells by inducing membrane protrusion [7,8]. Because the process of actin cytoskeleton remodeling is also upregulated during metastasis, a therapeutic strategy targeting the cytoskeleton polymerization pathway could effectively block cancer progression by reducing cancer cell migration, the first step of metastasis [9].

In particular, the actin-related protein 2/3 complex (Arp2/3), which is involved in Arp2, Arp3, and ARPC1-5, plays a crucial role during cancer cell migration by regulating actin polymerization [10,11]. Since cell motility is modulated by lamellipodia, which are membrane protrusions at the leading edge, the Arp2/3 complexes are correlated with the malignancy of cancer [12,13]. Within the Arp2/3 complex, ARPC2 binds to actin filaments and helps with the approach of other Arp2/3 subunits [14,15]. Hetrick et al. reported that the Arp2 or Arp3 inhibitors CK666 and CK869 markedly inhibited the migration of cancer cells but also affected normal cells [16,17]. However, we have previously proven that ARPC2 inhibitors block cancer cell migration with no effect on normal cells [16,18]. Thus, ARPC2 inhibitors may be good prospective therapeutic agents for cancer treatment without toxicity to normal cells.

Benproperine (Benp), which is well known as a cough suppressant, is a racemic mixture of R- and S-Benp. Chen et al. revealed that enantiomers of Benp do not show any differences in antitussive effects [19]. Recently, Benp was discovered to have an antimetastatic effect by inhibiting ARPC2 [18], but the effect of its isoforms on cancer cell migration has not been examined.

In this study, we investigated whether stereoisomers of Benp inhibit the migration of cancer cells differently. We evaluated the cancer-specific antimigration effect of Benp isomers based on Transwell migration and invasion assays. The metastatic inhibitory effect of S-Benp was assessed in a mouse animal model. We determined the binding affinity of S-Benp to ARPC2 by surface plasmon resonance analysis [20,21] and used label-free nonaffinity target identification approaches, such as the drug affinity responsive target stability (DARTS) and cellular thermal shift assay (CETSA), to further validate ARPC2 as a direct target of S-Benp [16,18]. These techniques have recently been widely used as a method to verify the binding between drugs and target molecules. The binding site of S-Benp in ARPC2 was also confirmed by CETSA and DARTS in ARPC2 point-mutated cells [22,23]. Using Cy3-conjugated S-Benp and CETSA, we also observed that S-Benp effectively localized and bound to ARPC2 in vitro and in vivo. Furthermore, the lamellipodia formation inhibitory effect of S-Benp was elucidated through an actin polymerization assay and immunofluorescence imaging. This study shows the therapeutic possibility of S-Benp and provides a clue for antimetastatic drug discovery by targeting ARPC2.

## 2. Results

### 2.1. Antimetastatic Effect of S-Benproperine

In a previous study, we confirmed that 20 µM Benp treatment reduced cell viability by 20% in various cell lines [18]. Based on the previous results, we investigated the proliferation inhibitory effect of Benp isomers in cancer cells (Figure 1A,B). DLD-1 cells were treated with 0, 1, 2, 5, 10, and 20 µM R- or S-Benp for 24, 48, and 72 h. R- or S-Benp treatment for 2 days did not affect cell viability; however, treatment for 3 days induced approximately 20% growth inhibition in DLD-1 cells. Then, we examined the migration inhibitory effect of Benp, R-, and S-Benp in a couple of cancer cells. Interestingly, S-Benp strongly inhibited the migration of cancer cells compared with R-Benp (Figure 1C,D). Both Benp and S-Benp (5 µM) significantly blocked DLD-1, AsPC-1, and B16-BL6 cancer cell migration, contrary to R-Benp-treated cells. In particular, the inhibitory effect of S-Benp on DLD-1-cell migration was more than twice as strong as that of Benp treatment (Figure 1C). However, none of the isomers of Benp affected the migration of MCF-10A (mammary gland epithelial cells, Figure 1E) up to 10 µM.

Next, we compared the antimetastatic effects of Benp and S-Benp in DLD-1 cells (Figure 2A,B) through migration and invasion assays.

When DLD-1 cells were treated with Benp at 1 µM or 2 µM, migration was inhibited by 18% and 53%, respectively, while S-Benp blocked migration by 52% and 78%, respectively. Interestingly, when cells were treated by S-Benp with a concentration as low as 0.5 µM, cell migration was inhibited 10 times more strongly than when treated with Benp at the same concentration. Benp and S-Benp effectively suppressed cancer cell migration with IC_50_ values of 2 µM and 1 µM, respectively. Additionally, 2-µM or 5-µM S-Benp treatment hampered cancer invasion by 57% and 93%, respectively. Benp has relatively lower anti-invasive activity than S-Benp, and Benp, and S-Benp inhibited the invasion of DLD-1 cells with IC_50_ values of approximately 4 µM and 2 µM, respectively. Taken together, S-Benp is an active stereoisomer of Benp and effectively decreased the migration and invasion of cancer cells compared with Benp and R-Benp treatment.

Based on the migration inhibition by S-Benp, we determined whether S-Benp suppressed tumor metastasis using luciferase-expressing pancreatic cancer AsPC-1 cells. After luciferase-expressing AsPC-1 cells were injected into the pancreas of BALB/c nude mice, orthotopic tumor growth was monitored for 17 days by bioluminescence imaging to measure luciferase activity. S-Benp strongly inhibited the growth of primary pancreatic tumors compared to vehicle control (50.8% by tumor weight and 71.6% by photon flux of the full body) with no change in body weight (Figure 3A–C). As shown in Figure 3D, S-Benp significantly suppressed pancreatic tumor cell metastasis into the liver, spleen, and kidney compared to vehicle control by 40.3%, 55.5%, and 88.3% inhibition, respectively, and strongly reduced luciferase expression in the pancreas (63.6%). These data suggest that S-Benp significantly inhibited cancer cell migration and suppressed pancreatic cancer metastasis into the major organs.

### 2.2. S-Benproperine Directly Binds to ARPC2

In a previous study, we revealed that Benp inhibited cancer cell migration by binding to ARPC2 [18]. To determine the difference in ARPC2 binding affinity between R- and S-Benp, we conducted surface plasmon resonance (SPR) analysis, a cellular thermal shift assay (CETSA), and drug affinity responsive target stability (DARTS). SPR analysis is generally used for calculating the kinetic constant of molecular binding interactions, including protein–drug, nucleic acid–protein, and protein‒protein interactions [20,21]. An interaction between S-Benp and ARPC2 was detected in a dose-dependent manner, while R-Benp was nonspecifically associated (Figure 4A). The equilibrium dissociation constant (KD) for S-Benp and ARPC2 was 1.12 × 10^−6^ M^−1^. However, in the case of R-Benp, we could not measure KD values because of nonspecific interactions between R-Benp and ARPC2.

Additionally, we performed nonlabelled biochemical methods, such as CETSA and DARTS, to validate the direct engagement between the drug and targeting proteins [22,23]. It was confirmed that ARPC2 was stabilized more than 2-fold against heat-induced denature in the presence of 100 µM S-Benp according to CETSA (Figure 4B). Additionally, 100 µM S-Benp protected against approximately 2-fold ARPC2 digestion by pronase compared to the control (Figure 4C). However, R-Benp did not induce any resistance to heat or protease of ARPC2. Collectively, these data indicated that S-Benp directly binds to ARPC2, while R-Benp only weakly fluctuates binding to ARPC2.

### 2.3. Validation of Binding Sites of S-Benproperine in ARPC2 Using Mutated Cancer Cells

In a previous study, we confirmed the binding pocket of Benp in ARPC2 through a molecular docking study, coimmunoprecipitation assays, and DARTS [18]. To elucidate the binding pocket of S-Benp in ARPC2, we conducted migration assays, CETSA, and DARTS using cells with the F225A mutant of ARPC2. The cells were transfected with the ARPC2 mutant after the elimination of endogenous ARPC2 expression, and the expression levels were confirmed in a previous study [16]. As shown in Figure 5A, 49.7% of the migration inhibitory effect of S-Benp in the cells transfected with wild-type ARPC2 was reduced by up to 10.6% in ARPC2^F225A^ cells. Next, CETSA and DARTS experiments were conducted to prove whether the decrease in the activity in ARPC2^F225A^ cells was caused by weak binding between S-Benp and ARPC2^F225A^.

Interestingly, S-Benp did not affect the thermal stability of ARPC2^F225A^ but increased the stability of ARPC2 WT by two-fold (Figure 5B). Furthermore, 100 µM S-Benp increased the proteolysis resistance of ARPC2 WT cells by approximately two-fold, contrary to ARPC2^F225A^ (Figure 5C). Therefore, these results show that S-Benp inhibits cancer cell migration by binding to the reported binding site of Benp [18].

### 2.4. S-Benproperine Localization and Binding to ARPC2 In Vitro and In Vivo

To investigate whether S-Benp could reach and bind to ARPC2 in cancer cells and tumor tissues, we examined the intracellular localization of S-Benp using Cy3 fluorescence molecules and confocal microscopy. We synthesized Cy3-conjugated S-Benp (Cy3-S-Benp) to visualize the location of S-Benp (Figure 6A). With Cy3-S-Benp, we evaluated the viability and migration inhibitory effect using AsPC-1 cells. Although 10 μM Cy3-S-Benp treatment drastically suppressed cell viability and migration, Cy3-Benp (2 μM) inhibited AsPC-1-cell migration by 33.7% compared to the control group without affecting cell viability (Figure 6B,C).

Based on the results (Figure 6B,C), we examined whether Cy3-S-Benp was delivered and bound to intracellular ARPC2. On a fibronectin-coated plate, the Cy3 signal was observed in the cytoplasm and colocalized with ARPC2, showing that Cy3-Benp was directly bound to ARPC2 in intact cells (Figure 6D). To reveal the association between ARPC2 and Benp in vivo, we examined AsPC-1 tumor tissues extracted from BALB/c mice that were intravenously injected with Cy3-Benp (*n* = 2; Benp 5 mg/kg, single intravenous injection). Consistent with the results obtained in AsPC-1 cells, a significant increase in the Cy3 signal was apparent in the murine tumor tissues at 3 or 6 h after Cy3-Benp injection but not after free Cy3 injection. In addition, the Cy3 signal was mostly colocalized with ARPC2 within the AsPC-1 tumor tissues obtained from the Cy3-Benp-injected mice (Figure 6E). Collectively, Cy3-labeled Benp directly binds to ARPC2 in cultured cells in vitro and in tumor tissue in vivo. Moreover, we applied CETSA to AsPC-1 tumor tissue samples, which were obtained from AsPC-1 tumor-bearing mice at 3 or 6 h following the intravenous injection of S-Benp. S-Benp increased the thermal stability of ARPC2 in a time-dependent manner (Figure 6F,G). This thermal stability was distinctly increased by S-Benp treatment in a dose-dependent manner at 67 °C (Figure 6G). Taken together, our results suggest that S-Benp was effectively localized and bound to intracellular ARPC2 both in vitro and in vivo.

### 2.5. S-Benproperine Delayed Actin Polymerization

The Arp2/3 complex induces branched actin nucleation at the leading edge in migrating cells [24]. Thus, to evaluate whether R- or S-Benp treatment affects the actin polymerization rate, we performed an in vitro actin polymerization assay, which measured the fluorescence of pyrene-conjugated actin during the polymerization process [25]. The addition of S-Benp (50 µM) to a mixture of actin monomer, Arp2/3 complex, and WASP-VCA significantly delayed action polymerization, while R-Benp only slightly disturbed it (Figure 7A).

Additionally, the lamellipodia formation inhibitory effect of S-Benp was visualized through phalloidin staining [26] (Figure 7B). S-Benp treatment of DLD-1 cancer cells efficiently hindered polymerized actin at the cell periphery, leading to the inhibition of lamellipodia formation (Figure 7B). Vinculin, an actin-binding focal adhesion protein, plays a crucial role in the generation of traction forces as well as directionality in cellular migration [27,28]. The interaction of vinculin and actin also regulates Arp2/3 complex-mediated lamellipodia formation [29].

Thus, we observed the localization of actin and vinculin in S-Benp-treated cells using a fluorescence microscope. In Figure 7C, S-Benp treatment markedly reduced the interaction between actin and vinculin in comparison with control or R-Benp-treated cells. These data suggest that S-Benp efficiently impeded the initiation of action polymerization and lamellipodia formation by targeting ARPC2 function, which led to inhibition of the migration and metastasis of cancer cells.

## 3. Discussion

Since metastasis is a major cause of mortality and cancer therapeutic failure, it is important to develop new drugs targeting metastasis [30]. The cancer cells in primary tumors spread through migration and invasion as the initial process of metastasis [4]. Thus, targeting the migration inhibition process to block metastasis is a promising strategy for cancer treatment. In a previous study, we found that Benp suppressed cancer cell migration through ARPC2 inhibition [18]. However, stereoisomers of drugs might be different from their biological target or pharmacological properties, such as metabolism and absorption and the effect of their isomers [31,32]. Therefore, it is very important to identify an active stereoisomer of Benp, which is a racemic mixture. In this study, we found that S-Benp treatment suppressed the migration and metastasis of various cancer cells, contrary to normal cells (Figure 1). S-Benp strongly inhibited the migration and invasion of DLD-1 cells compared to R-Benp (Figure 2).

Mori et al. indicated that drug stereoisomers could affect the affinity of the binding pocket [33]. Thus, we clarified the difference in binding affinity to ARPC2 between R- and S-Benp (Figure 4). By SPR analysis, CETSA, and DARTS, we confirmed that S-Benp specifically bound the ARPC2 protein, contrary to that in R-Benp. In various cancers, including breast, gastric, and pancreatic cancer, it has been indicated that ARPC2 is correlated with cancer proliferation and migration and a low survival rate of patients [14,15,34]. Cheng et al. suggested that ARPC2 induced lymph node invasion and epithelial–mesenchymal transition (EMT). However, ARPC2, which regulates actin polymerization, is essential for cells since it is related to cell migration as well as cell division [15]. Modulating ARPC2 activity is important rather than eliminating ARPC2 expression as blocking ARPC2 expression also affects the expression of other Arp2/3 complexes [16]. Thus, our results suggest that S-Benp could be an efficient promising antimigration reagent by regulating ARPC2 activity. Furthermore, through the point mutation of ARPC2, we revealed the binding pocket of S-Benp (Figure 5). In a previous study, we predicted that the benzyl-phenyl moiety of Benp hydrophobically interacts with F225 in ARPC2 and confirmed this via a coimmunoprecipitation assay and DARTS [18]. We also confirmed that the binding site of S-Benp and Benp was the same using point-mutated cells. The target engagement was further confirmed by employing CETSA and DARTS, new methods that directly assess target protein–inhibitor interactions in cells. As shown in Figure 5, S-Benp inhibited cancer cell migration and stabilized ARPC2 against thermal and proteolytic degradation in ARPC2 WT cells but not in ARPC2^F225A^ cells. Pimozide, reported as an ARPC2 inhibitor, bound ARPC2 F225 and F247 through hydrophobic interactions [16]. However, movement of the bis-(4-fluorophenyl) group of pimozide to the deep binding pocket increased migration activity in ARPC2^F247A^ and ARPC2^Y250F^ cells [16]. In particular, the region around F225 in ARPC2 is an interaction site with ARP3, which initiates actin nucleation networks [18,35]. As ARPC2 generates a dimer with ARPC4, it provides the primary site of Arp2/3 complex binding to F-actin [36]. Thus, S-Benp might specifically bind to ARPC2 F225 and effectively inhibit the Arp2/3 complex by modulating the interactions of ARPC2 and Arp3.

Specific drug engagement with target molecules is important for ideal drug efficacy and few side effects [37]. In Figure 6, we confirmed that S-Benp was localized and bound to ARPC2 in vitro and in vivo. S-Benp treatment captured cytoplasmic ARPC2 and modulated its function. In addition to the analysis of S-Benp binding and colocalization with ARPC2, we verified that ARPC2 activity was significantly inhibited by only S-Benp treatment (Figure 7). During cell migration, front–rear cell polarity leads to changes in membrane tension and cell adhesion [38,39]. In the initial step of migration, actin polymerization at the leading edge of migrating cells provokes cells to move and form lamellipodia [40,41]. Binding of vinculin and actin filaments generates traction forces and changes cell spreading and cell adhesion [42]. Furthermore, branched actin by the Arp2/3 complex enhanced the upstream pathway of the Arp2/3 complex, such as vinculin and WAVE2, during directional migration [43]. F-actin bundles directly initiate lamellipodia formation via activation of adhesion signaling [44]. In particular, S-Benp disturbed lamellipodia formation and the colocalization of vinculin and F-actin in cancer cells. Therefore, our data indicate that S-Benp markedly diminished cancer cell migration metastasis by inhibiting ARPC2-mediated lamellipodia formation.

In conclusion, S-Benp is an active stereoisomer, and the biological activities of S-Benp occur by direct interaction with ARPC2 in cancer cells and tumor tissues. Interestingly, it was reported that when Benp was orally administered to humans, there was twice as much S-Benp in the blood as R-Benp [45]. Collectively, S-Benp may be a promising candidate for use as an antimigratory reagent and provides an effective therapeutic strategy for improving the lifespan of cancer patients.

## 4. Materials and Methods

### 4.1. Cell Lines and Cell Proliferation Assay

All cancer cells were purchased from the American Type Culture Collection (Rockville, MD, USA). DLD-1 (colon cancer cells), AsPC-1 (pancreatic cancer cells), and B16-BL6 (melanoma) cells were cultured in RPMI medium. MCF-10A cells were cultured in DMEM. All culture media were supplemented with 10% (*v*/*v*) FBS and 1% (*w*/*v*) penicillin‒streptomycin, and all cells were cultured at 37 °C with 5% (*v*/*v*) CO_2_.

Cells (5 × 10^3^ cells/well) were seeded in 96-well plates and incubated for 24 h before drug treatment for 24, 48, and 72 h. At the end of each time point, 10 µL of cell proliferation reagent WST-1 solution (Roche Diagnostics, Indianapolis, IN, USA) was added to each well, and the plates were incubated for 90 min at 37 °C. Cell viability was quantified by measuring the absorbance of the solution at 450 nm using a microplate reader (Bio-Rad, Hercules, CA, USA). The assay was performed in triplicate.

### 4.2. Transwell Migration and Invasion Assay

The migration assay was performed using Transwell inserts with an 8.0 µm pore size (BD Biosciences, San Jose, CA, USA). For the invasion assay, the inserts were coated with Matrigel basement membrane matrix (Corning Inc., Corning, NY, USA) for 1 h on a clean bench. After washing with serum-free medium, the inserts were used for the assay.

The starved cells were seeded into the upper chamber with serum-free medium. Each concentration of Benp isomers in 10% FBS medium was applied to the lower chamber. After 12–24 h, the migrated cells were stained with crystal violet (Sigma-Aldrich, St. Louis, MO, USA) for 10 min. The membrane was washed with PBS, and then, the cells were analyzed under a light microscope (Nikon Eclipse TE300; Nikon, Tokyo, Japan).

### 4.3. In Vivo Xenograft and In Vivo Metastasis Assay

All animal work was performed in accordance with a protocol approved by the Institutional Animal Care and Use Committee. For the in vivo orthotopic xenograft assay, AsPC-1 cells (9 × 10^5^ cells/mouse) that stably expressed luciferase were directly injected into the pancreas of female BALB/c nude mice (6 weeks old; Nara Biotech, Seoul, Korea). The luciferase activity in the mice was immediately imaged after injection to confirm that the cancer cells were successfully implanted. S-Benp was orally administered 5 days per week (50 mg/kg) from day 1 (the day after cancer cell injection). Tumor growth was monitored under a live animal imaging system (PHOTONE IMAGER, Biospace, Nesles-la-Vallée, FRANCE) after injecting luciferin (15 mg/mL, Gold Bio, St. Louis, MO, USA). On day 21, the mice were sacrificed and evaluated for metastatic tumor growth in major organs, such as the pancreas, liver, spleen, and kidney.

For drug delivery investigation, AsPC-1 cells (9 × 10^6^ cells/mouse) were subcutaneously injected into the right flanks of nude mice (6-week-old female BALB/c mice). A single dose of Cy3 (Lumiprobe) or Cy3-S-Benp was intravenously administered at 5 mg/kg when the average tumor volume reached 267.2 ± 18.1 mm^3^. Tumors were collected at 3 and 6 h after treatment for immunohistochemistry and the cellular thermal shift assay.

### 4.4. Purification of the Recombinant Protein

The ARPC2 forward primer (5**′**-CGGGATCCATGATCCTGCTGGAGGTGAACAACCG-3**′**) and ARPC2 reverse primer (5**′**-CGGAATTCCGGCGGGATGAAAACGTCTTCCCC-3**′**), containing BamHI and EcoRI restriction sites, respectively, were used to obtain the inserted genes to purify ARPC2 proteins. The PCR products were digested with BamHI and EcoRI and then inserted into the pET-28a protein expression vector. After transformation into the Escherichia coli BL21 strain, the expression of each protein was induced by IPTG and purified with a His-tag purification kit (Millipore).

### 4.5. Surface Plasmon Resonance Analysis

To determine the dissociation constant of compounds toward ARPC2, surface plasmon resonance experiments were performed with a Biacore T100 instrument (GE Healthcare Biosciences, Pittsburgh, PA, USA). ARPC2 was immobilized on the CM5 sensor chip (7600 RU) activated by a mixture of 1-ethyl-3-(3-dimethylaminopropyl)-carbodiimide and N- hydroxysuccinimide. Various concentrations of the compounds ranging from 250 nM to 4 μM were injected over protein surfaces for 90 s, and dissociations of the compounds were observed for 180 s at a flow rate of 30 μL/min at 25 °C. Tris (20 mM), 150 mM NaCl, 0.005% P20, and 2% DMSO were used as the running buffer. No surface regeneration was employed. Data were analyzed with Biacore T100 Evaluation software (GE Healthcare, Pittsburgh, PA, USA) by fitting the sensorgrams to the 1:1 binding model.

### 4.6. Immunoblotting

Total cell lysates were prepared by lysing cells in RIPA buffer (50 mM Tris, pH 7.0, 150 mM NaCl, 5 mM EDTA, 1% deoxycholic acid, 0.1% SDS, 30 mM Na_2_HPO_4_, 50 mM NaF, and 1 mM Na_3_VO_4_) containing a protease inhibitor cocktail (Roche Diagnostics). Lysates were placed on ice for 10 min and centrifuged for 10 min (13,000× *g*, 4 °C). Denatured proteins (20–50 μg) were separated using 12% sodium dodecyl sulfate‒polyacrylamide gel electrophoresis and transferred onto a polyvinylidene fluoride membrane (EMD Millipore, Billerica, MA, USA). The membrane was blocked for 1 h using 5% (*w*/*v*) skimmed milk in Tris-buffered saline containing Tween-20 (TBST) and incubated with the following antibodies: ARPC2 (Abcam, Cambridge, UK, ab133315) and glyceraldehyde-3-phosphate dehydrogenase (GAPDH) (Santa Cruz Biotechnology, Santa Cruz, CA, USA, sc-47724). After washing thrice (10 min each) with TBST, the membrane was incubated with horseradish peroxidase-conjugated goat anti-mouse or rabbit anti-goat IgG (Santa Cruz Biotechnology) in TBST containing 5% (*w*/*v*) skimmed milk at room temperature for 1 h. The membrane was rinsed thrice (10 min each) with 0.1% (*v*/*v*) TBST. An enhanced chemiluminescence system (Luminata Forte Western HRP substrate (EMD Millipore)) was used to visualize the bands using the LAS 4000 mini (GE Healthcare Biosciences). Densitometry of the bands was performed using the MultiGauge program (Fuji Photo Film Co., Ltd., Tokyo, Japan).

### 4.7. Drug Affinity Responsive Target Stability (DARTS)

Cells were harvested by scrapping with M-PER buffer (Thermo Fisher Scientific Inc., Rockford, IL, USA) supplemented with 1 mmol/L NaF, 1× protease inhibitor cocktail, and 1 mmol/L Na_3_VO_4_. After centrifugation at 13,000× *g* for 10 min, 1 mg/mL proteins were diluted in TNC buffer (50 mM Tris-HCl, pH 8.0, 50 mM NaCl, and 10 mM CaCl_2_) (Sigma-Aldrich, St. Louis, MO, USA). The lysates were incubated with DMSO or drug for 1 h at room temperature. Following the incubation, 20 μg of proteins were proteolyzed in each concentration of pronase (Roche Diagnostics, Indianapolis, IN, USA, 10165921001) for 10 min at room temperature. To stop the enzyme activity, SDS loading buffer were added to lysates and the proteins were analyzed by Western blotting.

### 4.8. Cellular Thermal Shift Assay (CETSA)

Cells or tissues were lysed in lysis buffer (50 mM Tris-HCl, pH 7.5, 100 mM NaCl, 0.2% NP-40, 5% glycerol, 1.5 mM MgCl_2_, 25 mM NaF, 1 mM Na_3_VO_4_, and 1× protease inhibitor cocktail). After centrifugation, the lysates (1 mg/mL) were incubated with DMSO or drug for 1 h at room temperature. Twenty micrograms of protein were heated for 5 min at the indicated temperatures. The soluble proteins were separated by centrifugation (13,000× *g*, 20 min, 4 °C) and then analyzed by Western blotting.

### 4.9. In Vitro Actin Polymerization Assay

Assays were performed using the Actin Polymerization Biochem Kit (Cytoskeleton, Inc., Denver, CO, USA) according to the manufacturer’s protocol. Briefly, actin diluted in g-buffer was depolymerized to monomer for 1 h at room temperature. After centrifugation for 100,000× *g* for 1 h at 4 °C, the Arp2/3 complex (Cytoskeleton Inc., 10 nmol/L), WASP-VCA domain (Cytoskeleton Inc., 400 nmol/L), 0.2 mM ATP, or compounds were added to a 96-well black plate, and then total g-actin (2 μmol/L) was added to start the assay.

### 4.10. Immunocytochemistry and Immunohistochemistry

To prepare fibronectin-coated dishes, 35 mm high-μ dishes (Ibidi, Fitchburg, WI, USA) were treated with 10 μg/mL fibronectin (Sigma-Aldrich) for 1 h at room temperature. The fibronectin-coated dishes were washed once with PBS, and the cells (1.0 × 10^5^ cells) were replated with DMSO or Cy3-conjugated S-Benp (2 μM) for 1 h. After washing with PBS twice, the attached cells were fixed with 4% paraformaldehyde in PBS for 10 min at room temperature. The fixed cells were permeabilized with 0.2% Triton X-100 for 10 min and blocked with 1.0% BSA in PBS for 1 h. The cells were incubated with an anti-ARPC2 antibody (Abcam, ab133315) followed by an Alexa Fluor 647-conjugated donkey anti-rabbit IgG antibody (Thermo Fisher Scientific, A-31573). The nuclei were counterstained with 2 μg/mL DAPI (4**′**, 6**′**-diamino-2-phenylindole) (Santa Cruz Biotechnology, sc-3598) in PBS for 2 min.

Tumor tissues were isolated, frozen in liquid nitrogen-cooled 2-methylbutane, and embedded in TBS tissue freezing medium (Thermo Fisher Scientific Inc., Rockford, IL, USA). En-micrometer-thick sections were obtained on a cryostat (Microm HM550; Thermo Fisher Scientific Inc., Rockford, IL, USA) at −20 °C and placed on uncoated slides. The cryosections were fixed with 1.5% paraformaldehyde and incubated with an anti-ARPC2 antibody (Abcam, ab133315), followed by incubation with an Alexa-conjugated secondary antibody (Thermo Fisher Scientific Inc., Rockford, IL, USA). All images were acquired on a laser scanning confocal microscope (LSM 510 META, Carl Zeiss, Oberkochen, Germany) and analyzed with LSM Version 3.2 software (Carl Zeiss, Oberkochen, Germany).

### 4.11. Synthesis of S-Benp, R-Benp, and S-Benp-Cy3 

S-Benp or R-Benp was prepared from the reaction of 2-benzylphenol and R-propylene oxide or S-propylene oxide, respectively. The purity of S-Benp and R-Benp was confirmed by HPLC with chiral column (Column: IB-N5 250 × 4.6 mm 5 μm). (S)-1-(1-(2-Benzylphenoxy)propan-2-yl)piperidine (S-Benp: [a]^25^D **−** 121.8° (c 1, EtOH); ^1^H NMR (300 MHz) δ 7.29–7.07 (m, 7H), 6.87 (t, J = 7.2 Hz, 2H), 4.07 (dd, 1H, J = 9.3, 4.8 Hz), 4.00 (s, 2H), 3.85 (dd, 1H, J = 9.0, 6.6 Hz), 2.97 (m, 1H), 2.55 (m, 4H), 1.56 (m, 4H), 1.42 (m, 2H), 1.11 (d, 3H, J = 6.6 Hz); ^13^C NMR (75 MHz) δC 156.6, 141.0, 130.4, 129.5, 128.8, 128.1, 127.4, 125.7, 120.3, 111.1, 69.9, 58.9, 50.4, 36.1, 26.5, 24.7, 13.2. (R)-1-(1-(2-Benzylphenoxy)propan-2-yl)piperidine (R-Benp): [a]^25^D + 15.94° (c 1, EtOH); ^1^H NMR (400 MHz) δ 7.27–7.14 (m, 7H), 7.08 (dd, J = 7.2, 1.2 Hz, 1H), 6.87 (m, 2H), 4.07 (dd, 1H, J = 8.8, 4.4 Hz), 3.99 (s, 2H), 3.85 (dd, 1H, J = 9.2, 6.4 Hz), 2.97 (m, 1H), 2.55 (m, 4H), 1.55 (m, 4H), 1.41 (m, 2H), 1.11 (d, 3H, J = 6.8 Hz); ^13^C NMR (100 MHz) δC 156.6, 141.0, 130.48, 129.6, 128.8, 128.1, 127.4, 125.7, 120.3, 111.1, 69.9, 58.9, 50.4, 36.1, 26.5, 24.7, 13.2.

S-Benp-amine was prepared for the synthesis of Cy3-S-Benp. Cy3-S-Benp was synthesized by the coupling reaction of S-Benp-amine with Cy3 in the presence of EDC hydrochloride (Sigma-Aldrich). The reaction mixture was stirred for 24 h at room temperature. Then, the reaction solution was diluted with methylene chloride and water. The organic layer was dried over anhydrous MgSO_4_ and filtered through a celite-packed glass filter. The filtrate was concentrated in vacuo and purified with silica gel flash column chromatography to provide the desired product Cy3-S-Benp; ^1^H NMR (400 MHz, CDCl3) δ 8.45 (t, J = 13.6 Hz, 1H), 7.58 (brs, 1H), 7.05–7.43 (m, 13H), 6.85–6.94 (m, 3H), 6.71–6.79 (m, 2H), 4.12–4.26 (m, 8H), 4.03 (d, J = 16.0 Hz, 1H), 3.89 (d, J = 16.4 Hz, 1H), 3.68 (q, J = 6.4 Hz, 1H), 3.55 (brs, 1H), 2.93 (brs, 4H), 2.48 (t, J = 7.2 Hz, 2H), 1.65–2.06 (m, 22H), 1.51 (brs, 2H), 1.40 (d, J = 6.0 Hz, 1H), 1.14 (t, J = 7.2 Hz, 1H); ^13^C NMR (100 MHz, CDCl3) δ 174.1, 173.5, 156.3, 155.9, 150.9, 142.2, 141.8, 140.7, 140.5, 131.2, 129.3, 129.3, 129.1, 128.9, 128.8, 127.7, 127.1, 125.5, 125.4, 122.2, 122.2, 121.3, 120.5, 111.4, 111.2, 111.1, 111.1, 104.6, 103.9, 66.5, 60.6, 53.5, 50.7, 49.0, 49.0, 26.3, 44.5, 38.6, 36.1, 30.5, 28.2, 26.8, 24.0, 23.2, 21.2, 12.2, 11.7; LRMS(ESI^+^) m/z calcd for C_55_H_69_N_4_O_3_ [M]^+^ 821.54; found 821.54 [46].

### 4.12. Statistical Analyses

All measurements were performed at least in duplicate, and all values are expressed as the mean ± standard deviation. The results were processed via analysis of variance using the t test to assess the statistical significance of differences between groups. Values of *p* < 0.05, *p* < 0.01, and *p* < 0.001 are denoted by *, **, and ***, respectively.

## Figures and Tables

**Figure 1 pharmaceuticals-15-01462-f001:**
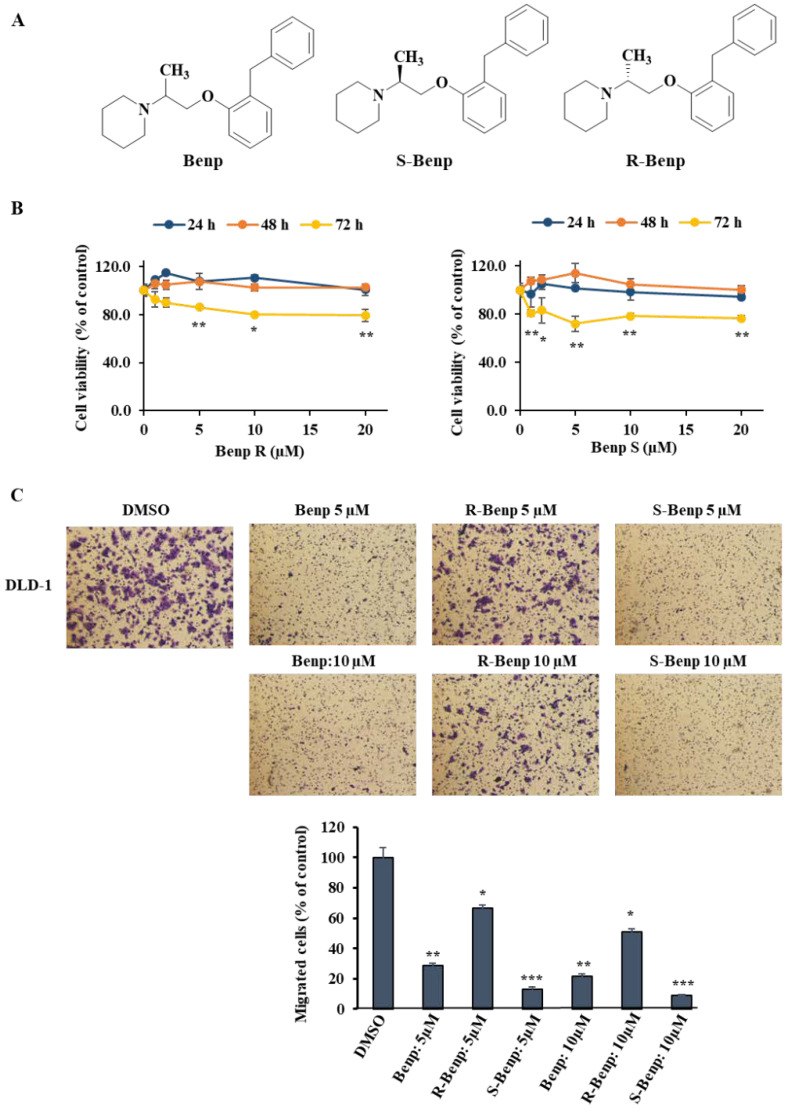
Inhibition of cancer cell migration by stereoisomers of benproperine. (**A**) Structure of isomeric benproperine (Benp). (**B**) Cytotoxic effect of R- and S-Benp. After treatment with R- or S-Benp for 24 h, DLD-1 live cells were measured via a WST-1 assay. (**C**) Migration inhibitory effect of Benp isomers in DLD-1 cells. (**D**) Migration inhibitory effect of Benp isomers in AsPC-1 and B16-BL6 cells. (**E**) Migration inhibitory effect of Benp isomers in MCF-10A cells. Migrated cells were detected via a Transwell migration system and visualized through crystal violet staining. The data represent means ± s.d.; comparisons were performed with *t tests* (two groups); ***** *p* < 0.05, ****** *p* < 0.01, ******* *p* < 0.001.

**Figure 2 pharmaceuticals-15-01462-f002:**
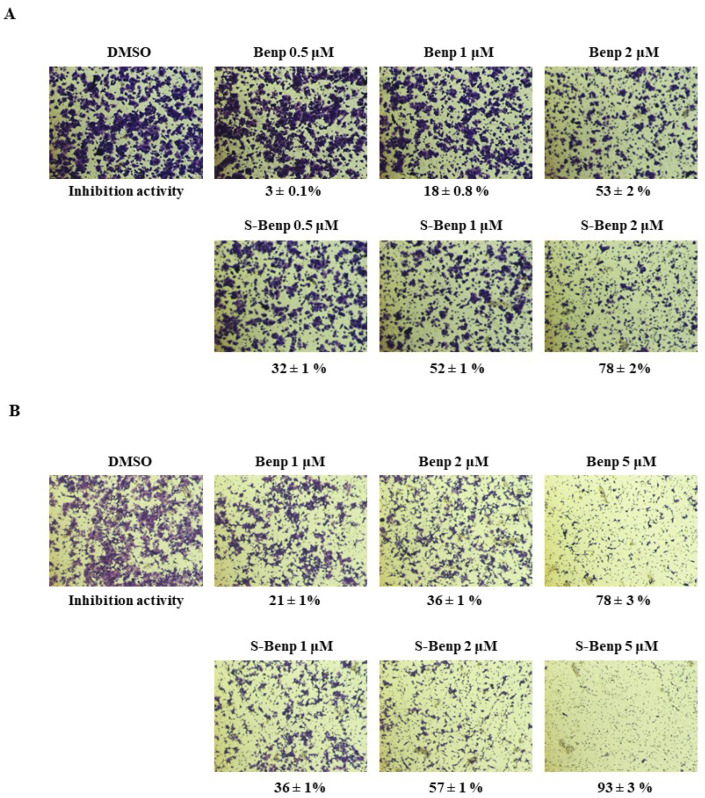
Benp and S-Benp inhibited cancer cell migration and invasion in a dose-dependent manner in DLD-1 cells. (**A**) Migration assay in Benp- or S-Benp-treated DLD-1 cells. Migrated cells were examined via a Transwell migration system and stained with crystal violet. (**B**) Cell invasion assay of Benp and S-Benp at 1, 2, and 5 µM in DLD-1 cells. Using Matrigel-coated Transwell inserts, the invasive cells were observed and analyzed. The data represent means ± s.d.; comparisons were performed with *t* tests (two groups).

**Figure 3 pharmaceuticals-15-01462-f003:**
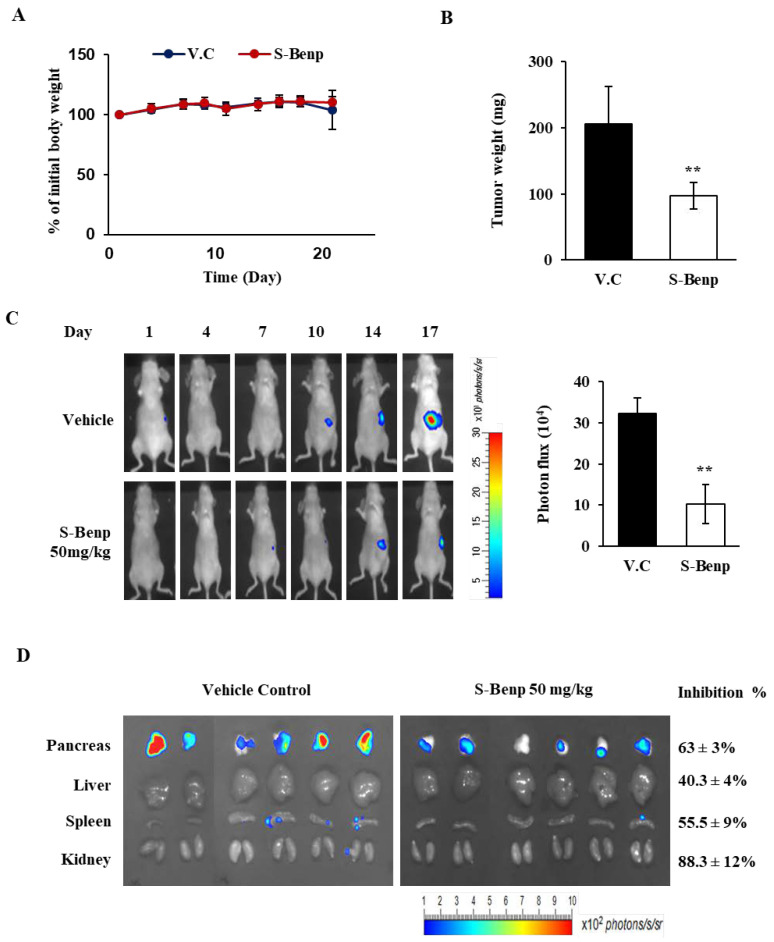
Inhibition of pancreatic cancer metastasis by S-Benp. (**A**) Change in body weight of vehicle- or S-Benp-treated mice over 21 days. (**B**) Reduction in tumor growth by S-Benp in a mouse model. (**C**) Representative images from luciferase-expressing AsPC-1 cells in the whole body. (**D**) Visualization of bioluminescence in the pancreas, liver, spleen, and kidney by luciferase-expressing AsPC-1 cells. The color scale indicates radiance (× 10^2^ photons/s/sr). The data represent means ± s.d.; ** *p* < 0.01.

**Figure 4 pharmaceuticals-15-01462-f004:**
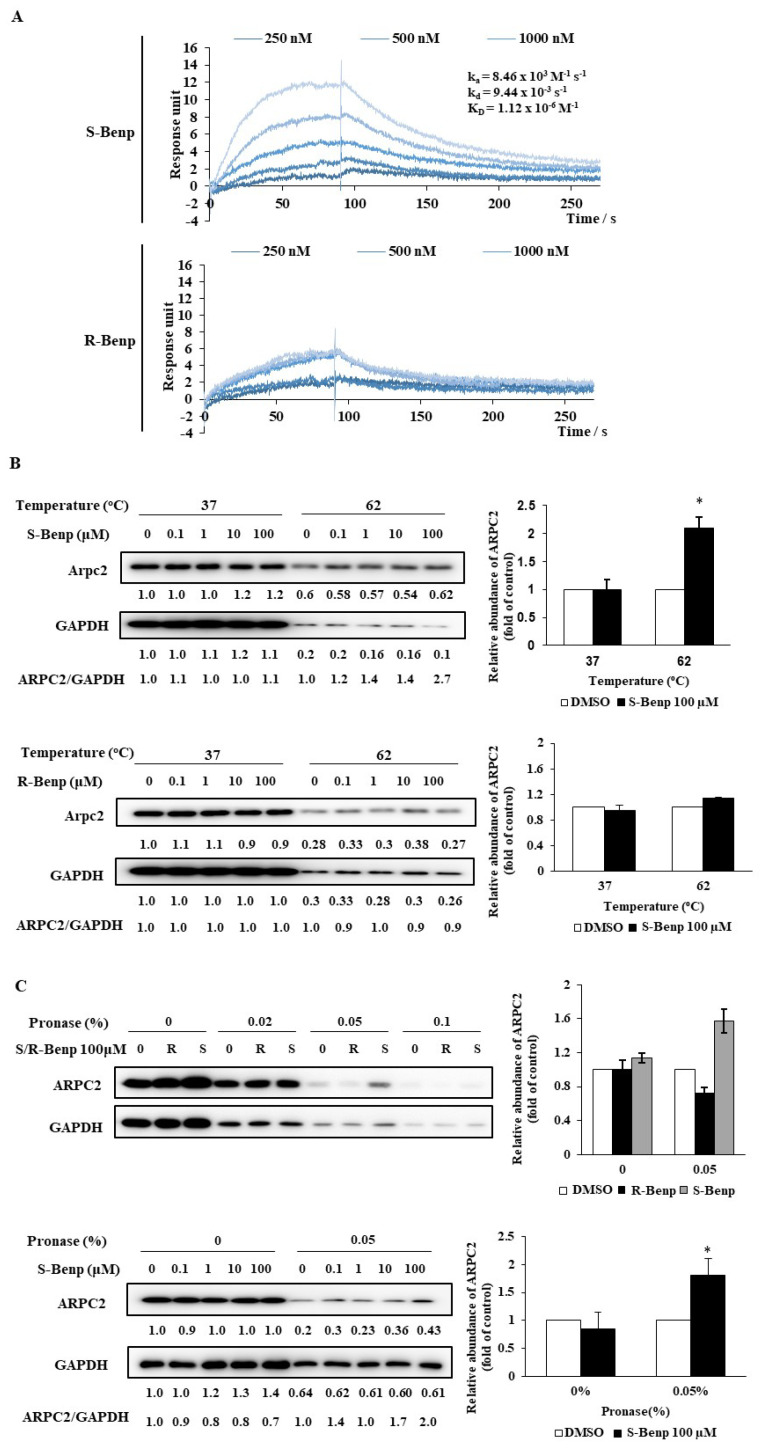
Direct binding of S-Benp with ARPC2. (**A**) Analysis of the interaction between ARPC2 and R- or S-Benp by SPR. After the immobilization of ARPC2 on the CM5 sensor chip, each concentration of R- or S-Benp was injected over the protein surface. Dissociation of R- or S-Benp was measured and analyzed by fitting the sensorgrams to the 1:1 binding model. (**B**) CETSA after incubation with DLD-1 cell lysates with DMSO or isoforms of Benp. (**C**) DLD-1 cell lysates were proteolyzed after incubation with DMSO or isoforms of Benp. The graph shows the quantification of the ARPC2 intensity with normalization against GAPDH (nontarget protein). The data represent means ± s.d.; comparisons were performed with *t* tests (two groups); ***** *p* < 0.05.

**Figure 5 pharmaceuticals-15-01462-f005:**
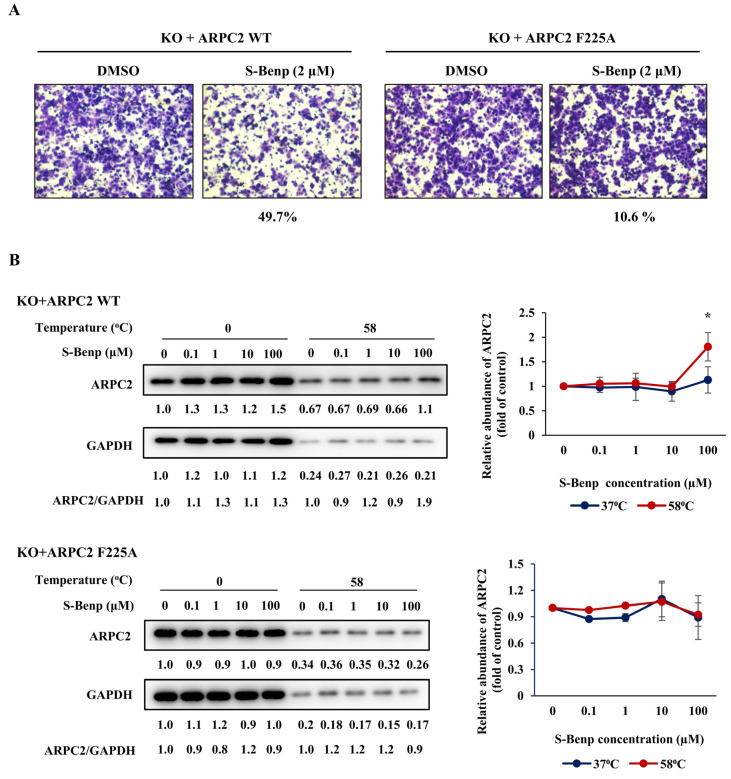
Identification of S-Benp binding sites using ARPC2-mutated cells. (**A**) Transwell migration assay of ARPC2−/− cells stably transfected with ARPC2 WT or ARPC2 F225A and treated with S-benproperine (S-Benp) for 18 h. The migrated cells were counted using the Image-ProPlus 5.0 program. (**B**) Immunoblotting of ARPC2 and GAPDH followed by incubation with S-Benp and ARPC2 WT or F225A cells after protein denaturation by heat. (**C**) DARTS after incubation with S-Benp and lysates of ARPC2 WT or F225A cells. The graph shows the quantification of the ARPC2 intensity with normalization against GAPDH (nontarget protein). The data represent means ± s.d.; comparisons were performed with *t* tests (two groups); ***** *p* < 0.05.

**Figure 6 pharmaceuticals-15-01462-f006:**
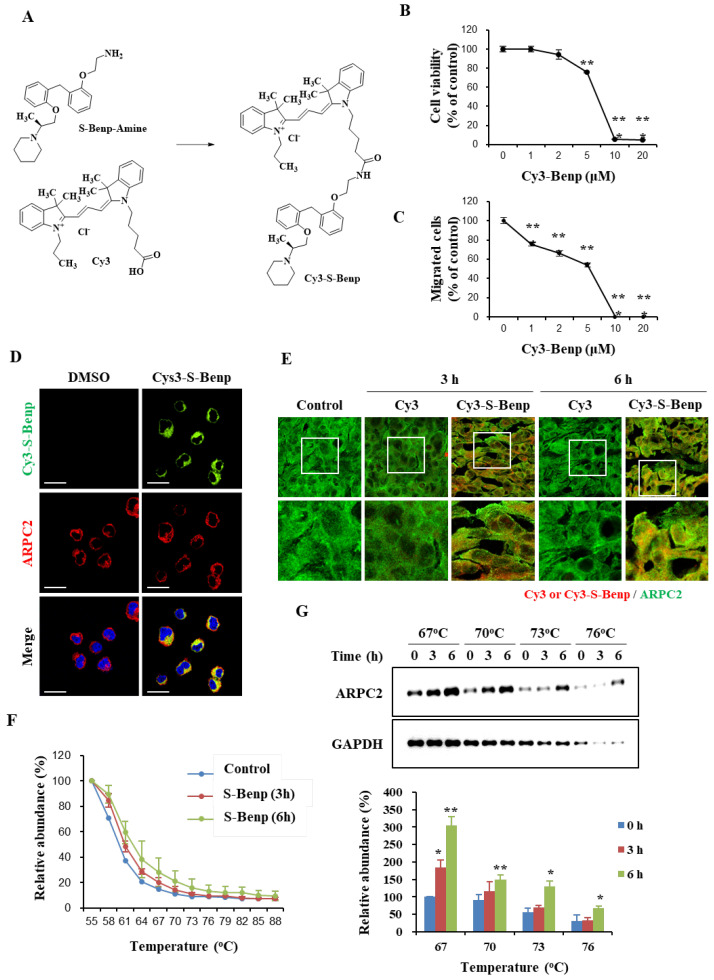
S-Benp localization and binding to ARPC2 in vitro and in vivo. (**A**) Synthesis of Cy3-S-conjugated benproperine (Cy3-S-Benp). (**B**) The viability of AsPC-1 cells that were treated with DMSO or Cy3-S-Benp for 24 h was measured using WST-1 (*n* = 3). (**C**) Cell migration assay of AsPC-1 cells that were treated with DMSO or various concentrations of Cy3-S-Benp for 24 h (*n* = 3). (**D**) Intracellular distribution of Cy3-S-Benp in AsPC-1 cells that were replated on fibronectin-coated plates for 1 h. The immunofluorescence analysis of ARPC2 following a 1 h treatment with DMSO or Cy3-S-Benp (2 μM) is shown (*n* = 2). Scale bars, 20 μm. (**E**) Colocalization of Cy3-S-Benp and ARPC2 in pancreatic tumor tissues at 3 or 6 h after intravenous injection of free Cy3 or Cy3-S-Benp (*n* = 2). (**F**,**G**) A cellular thermal shift assay was performed on tumor tissue lysates with increasing temperature (55 °C to 88 °C, interval temperature: 3 °C). Quantification of the soluble proteins immunoblotted for ARPC2 at the indicated temperature (*n* = 2 per group). The data represent means ± s.d.; comparisons were performed with *t* tests (two groups); ***** *p* < 0.05, ****** *p* < 0.01.

**Figure 7 pharmaceuticals-15-01462-f007:**
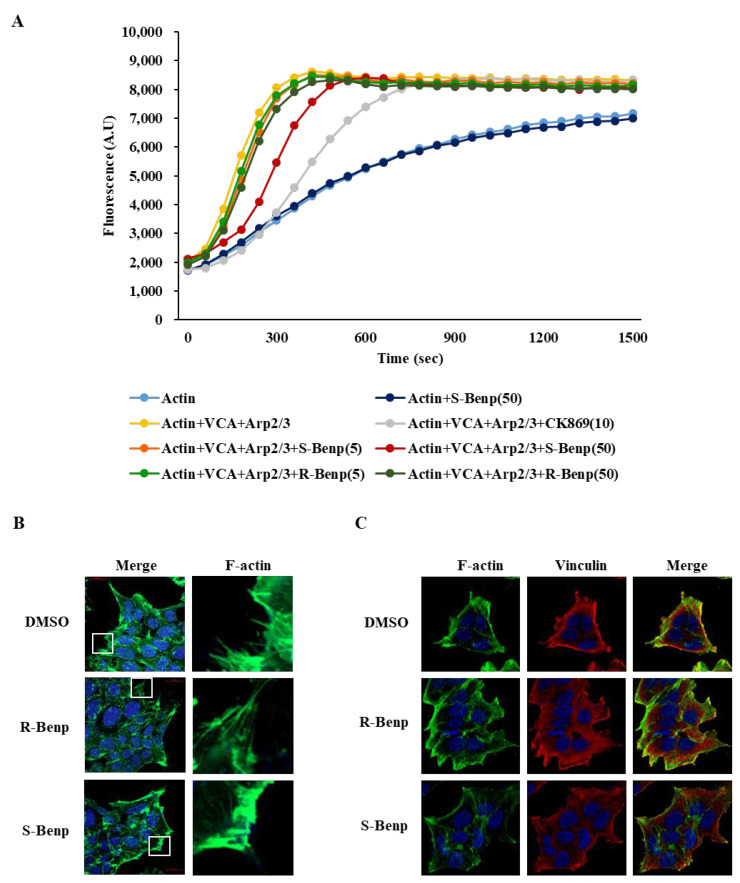
Inhibition of the initiation of actin polymerization and lamellipodia formation by S-Benp. (**A**) Arp2/3-dependent actin polymerizations with VCA with benproperine isomers or CK-869. Polymerized actin was measured by pyrene fluorescence at 365/405 nm (*n* = 2). (**B**,**C**) Confocal images of DLD-1 cells treated with DMSO or R- or S- Benp (2 μM) for 6 h.

## Data Availability

Data are contained within the article.

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
