# Peer review of "S-Benproperine, an Active Stereoisomer of Benproperine, Suppresses Cancer Migration and Tumor Metastasis by Targeting ARPC2"

_pharmaceuticals, 2022, doi:10.3390/ph15121462_

Round 1

Reviewer 1 Report

Authors present their study on S-Benproperine, an active stereoisomer of benproperine, which suppresses cancer migration and tumor metastasis by targeting ARPC2.

This work is mainly incremental development of their previous article describing that (racemic) benproperine targets ARPC2 (Biochem Pharmacol 2019, 163, 46-59, doi:10.1016/j.bcp.2019.01.017.)

Thus, this article is limited in novelty.

Methodology is in major part identical, only instead of benproperine, S-Benproperine is being used. There are also some methodological novelties as Cy3-conjugated S-Benp

Results comparing benproperine and S-Benproperine are expected and of no real novelty (authors do not answer some relevant questions, which would be needing information that R or S enantiomer is active).

Information is or course of great interest for somebody doing in the field, but merely as the justification, which single enantiomer authors used in all further studies. In this part article is highly limited.

I do not support the publication is such form.

Minor comments: enantiomer are not sufficiently evaluated, as enantiomeric purity is here of great importance. Authors should determine e.e. using more sensitive method than polarimetry, chiral chromatography would be suitable.

Synthesis should be described more in detail (regarding conditions) and this is again important regarding stereochemistry (possible racemization due to conditions). If authors used the same method they should cite US 2016/0095845 A1.

Reviewer 2 Report

The manuscript "S-Benproperine, an active stereoisomer of benproperine, suppresses cancer migration and tumor metastasis by targeting ARPC2" by H.-J. Jang et al. convincingly confirms the leading role of S-benproperine as an active stereoisomer in racemic benproperine, which was previously identified by the authors as an antimetastatic agent that inhibits ARPC2. The obtained results allow the authors to propose S-Benp as a potential drug for antimetastatic therapy.

The manuscript is well and logically organized. However, the authors must make a number of changes in order to make the manuscript accepted:

1. The most serious remark to the authors concerns the incorrect representation of the spatial structure of S- and R-stereoisomers in Fig. 1a and 6a.

2. Specify the name of the cells for the caption to Figure 1B (DLD-1 cells).

3. L. 106: "... when S-Benp was treated..." should be replaced by "... when cells was treated by S-Benp...".

4. Mark fragment "B" in Figure 6.

5. In both 13С NMR spectra, the number of signals (17 signals) does not correspond to the number of carbon atoms (21 atoms) in the isomeric structures. The shifts of the overlapping signals should be specified.

After corrections, the manuscript can be accepted for publication.

Round 2

Reviewer 1 Report

Authors have improved article according to the suggestions. I do not have additional comments.